# Semantic Object Navigation with Segmenting Decision Transformer

## Abstract

Understanding scene semantics plays an important role in solving the object navigation task, where an embodied intelligent agent has to find an object in the scene given its semantic category. This task can be divided into two stages: exploring the scene and reaching the found target. In this work, we consider the latter stage of reaching a given semantic goal. This stage is particularly sensitive to errors in the semantic understanding of the scene. To address this challenge, we propose a multimodal and multitasking method called SegDT, which is based on the joint training of a segmentation model and a decision transformer model. Our method aggregates information from multiple multimodal frames to predict the next action and the current segmentation mask of the target object. To optimize our model, we first performed a pre-training phase using a set of collected trajectories. In the second phase, online policy fine-tuning, we addressed the problems of long-term credit assignment and poor sampling efficiency of transformer models. Using the PPO algorithm, we simultaneously trained an RNN-based policy using ground-truth segmentation and transferred its knowledge to the proposed transformer-based model, which trains the segmentation in itself through an additional segmentation loss. We conducted extensive experiments in the Habitat Sim environment and demonstrated the advantage of the proposed method over the basic navigation approach as well as current state-of-the-art methods that do not consider the auxiliary task of improving the quality of the segmentation of the current frame during training.

## 1 Introduction

Navigating an intelligent agent (e.g. a robot) to a target object in an unknown environment is still a challenge for existing methods. This is confirmed by the results of modern benchmarks, for example in the simulators Habitat (Savva et al., 2019), AI2Thor (Kolve et al., 2017), and others. There are several reasons for this. First, the best existing neural network models that can operate in real time still do not segment objects reliably enough, especially when they are far away or partially visible (Miao et al., 2024; Kim et al., 2024). Second, the prediction of agent actions from visual data is also performed with a large number of errors and has significant improvement potential for both modular approaches (Chaplot et al., 2020) and end-to-end neural network models (Chen et al., 2023).

A separate problem is the related task of image sequence segmentation for intelligent agents. There are several approaches based on direct fusion of image sequence features (Shang & Ryoo, 2023; Su et al., 2023), auto-regressive prediction of segmentation masks based on previous masks and images (Šarić et al., 2021; Graber et al., 2022), consideration of three-dimensional constraints when segmenting objects of the sequence (Zhang et al., 2023b; WAN & FANG, 2023; Scarpellini et al., 2023), including those based on Gaussian blending (Zhu et al., 2024; Lei et al., 2024). However, in terms of the quality achieved, they still have significant limitations for use in the task of indoor navigation of an intelligent agent.

The navigation task is a partially observable reinforcement learning (RL) problem where history is fed into a sequence model (Sutton & Barto, 2018). While transformers are powerful tools in CV and NLP tasks (Brown et al., 2020; Zhang et al., 2024) and have long-term memory capability with effective representation learning from context for specific tasks (Lu et al., 2024), they generally have poor sampling efficiency and do not improve long-term credit assignment compared to recurrent

Figure 1: The Semantic Object Navigation task requires the agent to reach the target object, seen from the start position, within a distance of $1.0m$. Red dots on the map indicate areas where goal-type objects are located, and the resulting agent's path is indicated with the blue line.

neural networks (RNNs) (Ni et al., 2023). To overcome these limitations, we propose a method that simultaneously trains RNN-based and transformer-based versions of the policy. The advantage of this approach is that the RNN-based policy can effectively solve the navigation task by accessing the ground truth segmentation from the simulator, while the Transformer-based policy can predict the segmentation from the RGB sequence of frames and predict the sequence of actions by transferring knowledge from the RNN-based policy.

In this work, we propose to combine the action prediction of an intelligent agent and the task of RGB-D image sequence segmentation in a single transformer model. We will further show that such a solution allows us to improve the quality of image segmentation and action generation to solve the navigation problem for an object specified by a semantic label. Such semantic object navigation (see Fig. 1) can be useful in robotics applications where an embodied agent navigates in a non-deterministic environment (Batra et al., 2020a).

The main contributions of the article include the following:

- We developed a multimodal and multitask method called SegDT, which is based on training a single segmentation decision transformer model. The model aggregates information from multiple multimodal frames to predict the next action and the segmentation mask of the target object. Each frame consists of the current image, depth, target category, segmentation mask, and action.

- We proposed a two-phase training procedure for our module based on reinforcement learning. First, we performed a pre-training phase using a set of collected trajectories. In the second phase, online policy fine-tuning, we addressed the problems of long-term credit assignment and poor sampling efficiency of the transformer models. Using the PPO algorithm, we simultaneously trained an RNN-based policy using ground-truth segmentation and transferred its knowledge to the proposed transformer-based model, which trains the segmentation in itself through an additional segmentation loss.

- We conducted extensive experiments in the Habitat Sim environment and demonstrated the advantage of the proposed method over the basic navigation approach, as well as current state-of-the-art methods that do not consider the auxiliary task of improving the quality of the segmentation of the current frame during training.

## 2 RELATED WORK

Recent methods for object goal navigation use scene semantic information for action prediction to reduce overfitting and increase the navigation quality for unseen environments. The scene semantic can be available in the form of a 2D semantic segmentation mask. For instance, authors of the THDA method (Maksymets et al., 2021) introduce a policy network that uses depth and multichannel semantic masks as input. SkillFusion approach (Staroverov et al., 2023) proposes a goal-reaching policy that leverages an RGB observation and a binary segmentation mask of object goal. During inference time the success rate of such navigation approaches heavily relies on the quality of input

segmentation masks (Staroverov et al., 2023). Despite the active development of neural network architectures, the state-of-the-art methods for semantic segmentation (e.g. Mask2Former (Cheng et al., 2022), OneFormer (Jain et al., 2023), OpenSeeD (Zhang et al., 2023a), MQ-Former (Wang et al., 2024)) still show imperfect segmentation quality, especially for indoor environments, where objects can vary a lot within one semantic category.

In addition, the state-of-the-art methods for semantic segmentation do not take into account the peculiarities of an embodied agent interacting with its environment during navigation. The agent has a limited field of view, therefore instant observations may contain erroneous semantics when looking at the object from certain view angles. During the navigation episode, the agent can update its semantic understanding of the scene by observing the scene from more advantageous viewpoints. Such refinement can occur explicitly by using the accumulated semantic map of the environment (Tao et al., 2024; Morilla-Cabello et al., 2023). The explicit semantic maps of the environment can be used as input to predict action policy (Ramakrishnan et al., 2022; Zhang et al., 2023b; Yu et al., 2023). Other methods, such as (Chen et al., 2023), use implicit maps to model the history of observations. A major drawback of these methods is that as the navigable space expands, the size of the dense voxelized map can become infinitely large.

In contrast, we use a method that aggregates sequence information from previous semantic observations to refine semantic segmentation on the current frame and predict the next action. In this sense, our method is related to methods that solve the task of video segmentation (Zhang et al., 2023c; Shin et al., 2024). However, unlike such methods, our approach allows the agent to control its observations to navigate to the goal and improve the segmentation quality. At the same time, our method differs from existing embodied computer vision methods (Fan et al., 2023; Ding et al., 2023; Yang et al., 2019; Kotar & Mottaghi, 2022). These methods aim to improve the quality of visual perception, while our method increases both the quality of navigation and the quality of segmentation. The methods for embodied computer vision often operate in the next-best-view paradigm or use a small sequence of frames to predict the next action. However, the agent needs a longer history of observations to successfully solve the object goal navigation task. Unlike (Shang & Ryoo, 2023), we consider a complex photo-realistic 3D environment of the HM3DSem v0.2 (Yadav et al., 2023b) scenes.

A special feature of our method is the joint training of a semantic segmentation model and a transformer to predict the next actions. Previous works (Maksymets et al., 2021; Hong et al., 2023) consider semantic loss as an additional task for model training. However, these methods use semantic loss only to improve the action policy, and not to improve the quality of semantic segmentation by aggregating information from a sequence of frames.

## 3 TASK SETUP

In the literature (Batra et al., 2020b), the Semantic Object Navigation task is defined as follows. An agent is randomly initialized within an unfamiliar environment and needs to navigate toward an instance of a specified object category $C \in \{c_1, c_2, ..., c_n\}$ (e.g., a *plant*). The solution of this task usually consists of two stages. First, the agent explores the environment to find an instance of a given semantic goal. Next, the agent reaches the found object. In this work, we consider the second stage of reaching the semantic goal. Therefore, we initialized the agent at the random viewpoint of the semantic goal at a maximum distance of seven meters (Fig. 1).

Our problem can be formulated as a Partially-Observable Markov Decision Process (POMDP), defined as a tuple $(S, A, P, R, \rho_0, \gamma)$ for underlying observation space $S$, action space $A$, transition distribution $P$, reward function $R$, initial state distribution $\rho_0$, and discount factor $\gamma$.

In our setup, the agent receives an observation $S = (S_{RGBD}, C)$ at each step. We consider a discrete action space consisting of six types of actions: `callstop` to terminate the episode, `forward` by 0.25 m, `turnleft` or `turnright` by angle 15°, `lookup`, `lookdown` by turning the agent head by angle 30°. This type of discrete action space is common for indoor simulators such as Habitat Yadav et al. (2023b) or AI2-Thor Kolve et al. (2017).

The agent can take up to 64 steps in the environment. The episode finishes when the agent executes the `callstop` action. We assess the agent's performance via three common metrics for the Object

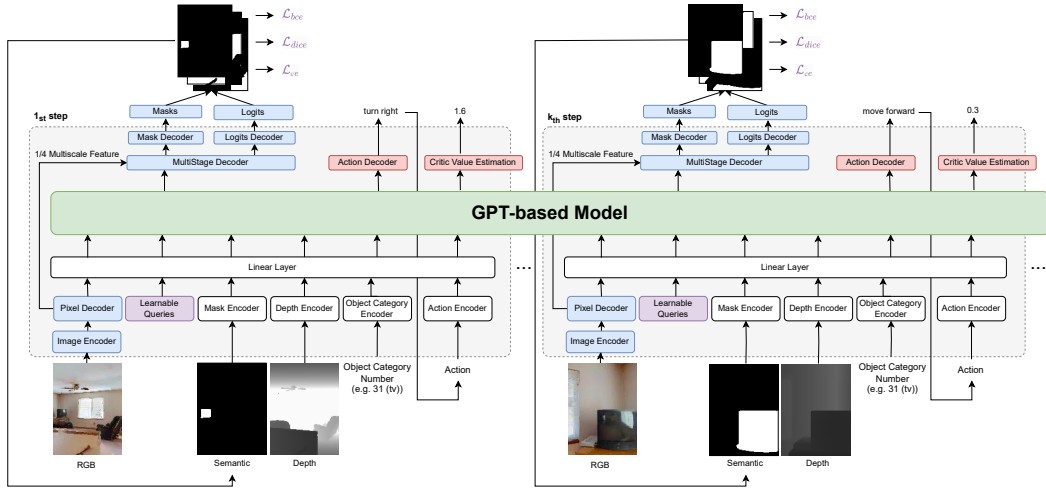

Figure 2: GPT architecture for predicting semantics and actions to complete the navigation task.

Navigation task Batra et al. (2020b): Success Rate (SR), Success weighted, i.e. inverse normalized, by Path Length (SPL), and SoftSPL.

## 4 METHOD

Our method consists of training a decision transformer model with a multistage mask decoder. Prediction at time $t$ involves two stages. An observation at time $t$ consists of an image $I_t$, a depth map $D_t$, and a target category name $c$. First, multi-scale feature maps of $I_t$ are generated using a ResNet50 backbone and a pixel decoder. These feature maps, along with trainable query features, are then fed into the decision transformer. After processing, the trainable query features are decoded by a multi-stage mask decoder to generate segmentation masks for a fixed set of categories. From the set of masks, a binary mask for the target category is selected and its embedding is extracted. This embedding, combined with the depth map and the category name embedding, completes the observation sequence embeddings. In the second step, the full sequence of observation embeddings is fed into the decision transformer to predict the probability distribution and state value of the next action. We then sample action $a_t$ and add its embedding to the observation sequence to predict actions at time $t + 1$. Figure 2 illustrates the model architecture.

### 4.1 SEGMENTATION MODULES

When choosing the architecture of the Segmenting Decision Transformer (SegDT) modules responsible for segmentation, we take Mask2Former (Cheng et al., 2022) as a basis. Mask2Former is one of the state methods for semantic segmentation. This method considers the segmentation problem as a problem of predicting a set of binary masks and their classification. The segmentation model is given an image of size $(H, W, C)$ as input.

The main components of Mask2Former are a backbone, a pixel decoder, and a multistage decoder. We use ResNet50 as the backbone. The output of the backbone is fed to the pixel decoder to generate 4 maps of high resolution per-pixel embeddings. The per-pixel embeddings have $1/4$, $1/8$, $1/16$, and $1/32$ of the resolution of the input image. We use a $1/32$ per-pixel embedding map as the image embedding for the Transformer model input.

In the original single-frame Mask2Former model, binary segmentation masks and their classification logits are decoded from $N$ learnable query features using multiscale feature maps. In our work, we use $N$ learnable query features as input to the Transformer model to take into account the context of previous observations. After passing through the transformer, the updated query features are passed through the multistage decoder. Here, similar to the Mask2Former model, we use multiscale feature maps to predict binary segmentation masks and their logits. From these binary masks,

a multi-channel semantic segmentation mask is formed for $N_{cl} = 40$. We then select the target semantic mask and use the ResNet50 encoder to create a semantic feature of size $(1, d_{sem})$. This feature describes the semantics of the current observation, similar to TDHA (Maksymets et al., 2021).

### 4.1.1 OBSERVATIONS EMBEDDINGS

For each time point, we describe the current observation using 29 embeddings obtained from different encoders and projected into the GPT hidden dimension $d_{GPT} = 768$. For each of the T-frames, we flatten the image pixel embeddings from Mask2Former into a sequence and project the image embeddings into $d_{GPT}$ using a linear layer. Thus, the image embedding for an image has a dimension of $(H \cdot W/32, d_{GPT})$. The learnable queries are represented by a set of 50 embeddings with dimension $(1, d_{GPT})$. We encode the semantics of each image using ResNet50 features obtained from the binary segmentation mask of the target object into a feature vector of dimension $(1, d_{sem})$. Thus, after projection, the embedding of semantic predictions for 1 image has a dimension of $(1, d_{GPT})$. We encode depth for each of the observations using ResNet18, resulting in a feature vector of dimension $(1, d_{depth})$. Using a linear layer, we project the depth features into the $d_{GPT}$ feature space. Thus, the feature embedding of the depth 1 observation has dimension $(1, d_{GPT})$. To encode the target category and the preformed action, we use a look-up table of learnable embeddings of dimensions $(N_{cl}, d_{GPT})$ and $(N_{actions}, d_{GPT})$, respectively. We populate the GPT input sequence with T observation embeddings. Thus, the dimension of the input sequence of observation embeddings is $(T \cdot (H \cdot W/32 + 4), d_{GPT})$.

### 4.1.2 PREDICTIONS

Since the goal of the semantic object navigation task is to reach an object of a certain target category, we expect that using the observation history can improve the segmentation quality for this target category. To decode semantic predictions, we use an idea from the original Mask2Former segmentation model (Cheng et al., 2022). We take the output learnable query features from the SegDT and pass them through the multistage decoder. To obtain the binary segmentation masks and their logits at time $t$, we additionally use the multi-scale feature maps predicted by the pixel decoder at time $t$. We use MLPs to decode the action distribution for the actor head and to estimate the state value for the critic head.

To predict the action at step t, we use the set of observations $\{o_0, ..., o_t\}$ and the previous actions $\{a_0, ..., a_{t-1}\}$. First, the sequence $\{o_0, a_0, ..., o_{t-1}, a_{t-1}, o_t\}$ is passed to the SegDT input to predict the segmentation masks $\{M_i^{pred}\}_{i=0}^t$. The mask corresponding to the target object category is used as the semantic observation for the time $t$. Next, SegDT makes another prediction of the action $a_t$, taking into account the segmentation mask, the depth, and the target category at time $t$. In this case, the last token of the output sequence of the transformer is used as input of the action decoder, i.e. the last token of the observation $o_t$.

## 4.2 LEARNING PROCESS

### 4.2.1 JOINT LEARNING ON OFFLINE DATA

As a central aspect of our experiment, we initialize the ResNet50 backbone, the pixel decoder, and the multi-stage decoder responsible for segmentation prediction with parameters of a pre-trained segmentation model. The primary goal during the initial phase of training is to establish an effective representation of the observations intended for navigation. To achieve this goal, we rely on an offline demonstration dataset composed of semantic goal-reaching instances between the start coordinates and the most proximal target. We collect the action probability distribution of a pre-trained RL agent with RNN and ground truth segmentation as input. During these initial stages, both SegDT (our multi-stage mask decoder) and our action decoder are trained simultaneously. To optimize mask prediction, we use the sum of the pixel-by-pixel binary cross-entropy $\mathcal{L}_{bce}$, the dice loss $\mathcal{L}_{dice}$, and the cross-entropy loss $\mathcal{L}_{ce}$ for mask classification as our loss function. Behavior cloning ($\mathcal{L}_{bce}$) is used to predict the action sequence. Additionally, we pretrain the Critic Value Decoder during this training phase. We use the pre-collected critic values obtained by the RL agent and apply an MSE loss $\mathcal{L}_{MSE}$ between them and the values predicted by SegDT, as articulated in equations 1 and 2.

$$\mathcal{L}_{total} = \lambda_{segm}\mathcal{L}_{segm}(\{\hat{M}^t, M^t, \hat{c}^t, c^t\}_{t=0}^T) + \lambda_{bce}^{act}\mathcal{L}_{bce}(\{\hat{a}^t, a^t\}_{t=0}^T) + \lambda_{MSE}\mathcal{L}_{MSE}(\{\hat{v}^t, v^t\}_{t=0}^T). \tag{1}$$

$$\mathcal{L}_{segm} = \lambda_{bce}^{segm}\mathcal{L}_{bce}(\{\hat{M}^t, M^t\}_{t=0}^T) + \lambda_{dice}\mathcal{L}_{dice}(\{\hat{M}^t, M^t\}_{t=0}^T) + \lambda_{ce}\mathcal{L}_{ce}(\{\hat{c}^t, c^t\}_{t=0}^T). \tag{2}$$

Here, $\hat{M}^t$ denotes the set of predicted masks at time $t$, $M^t$ are ground truth binary masks for object categories $c^t$, whereas $\hat{c}^t$ are predicted object categories. $\hat{a}^t, a^t, \hat{v}^t, v^t\}_{t=0}^T$ denote predicted action probability distribution, ground truth action probability distribution, predicted state value and ground truth state value respectively. $\lambda_{segm} = 1, \lambda_{bce}^{act} = 1, \lambda_{MSE} = 0.1, \lambda_{bce}^{segm} = 5, \lambda_{dice} = 5, \lambda_{ce} = 2$.

### 4.2.2 ONLINE FINETUNING

Limitations of Behavior Cloning arise primarily in two areas - an observable shift in distributions when confronted with different states at training and test times, and a lack of flexibility to adapt to evolving environments. In addition, when imitating suboptimal demonstrations, the results of Behavior Cloning subsequently reflect these imperfections. To overcome these limitations, in the second phase of policy training, we employed an online reinforcement learning approach that can incrementally adapt to changes in the environment.

However, online reinforcement learning (RL) requires a significant number of samples to achieve robust performance, which can be a significant limitation. In addition, the use of the transformer model introduces significant computational cost, especially for long sequences, as causal transformers require $O(t^2)$ time to compute the representation at time step $t$.

To address this issue, we sampled trajectories using an RNN-based policy that can be efficiently trained online with ground truth segmentation as input. Following the work of SkillFusion (Staroverov et al., 2023), we implemented the RNN-based GoalReacher skill. The limitation of this model is that it requires an external segmentation module during inference, which may output noisy segmentation masks that differ from those seen during training.

We then fine-tune the proposed SegDT policy with trajectories provided by the RNN-based policy. To generate actions, our model includes two segmentation-independent modules: actor and critic heads, similar to the RNN-based policy. To fine-tune these on SegDT, we transferred knowledge from the RNN-based policy using cross-entropy loss. In addition, we applied segmentation loss to SegDT and used PPO loss for both models (Fig. 3).

We demonstrate that integrating these insights into our pipeline significantly improves the performance of our navigation stack, achieving performance comparable to the advanced RNN-based policy with ground truth segmentation as input, without the need for an external segmentation module.

## 5 EXPERIMENTS

The main goal of our experiments is to navigate an autonomous agent toward its target object by minimizing cumulative distance and maximizing the understanding of the environment. To achieve this, we have followed a twofold training phase strategy: with the first phase focusing on obtaining high-quality semantic segmentation masks, and the second phase shifting towards action prediction with the use of an online Reinforcement Learning method for an adaptable learning experience.

### 5.1 EXPERIMENTAL SETUP

**Datasets.** The experiments were carried out in the Habitat environment (Savva et al., 2019). For the experiments, we select 146 training and 36 validation scenes of the HM3DSem v0.2 dataset (Yadav et al., 2023b). These scenes were divided into a training set of 173 scenes and a validation set of 9 scenes. Next, we sample episodes in each scene. The episode is characterized by the agent starting position, the coordinates, and the semantic type of the target object. We randomly sample starting points for episodes satisfying two conditions of the Goal Reaching task: the target object is in the agent's field of view and the agent is no more than 10 meters away from the goal.

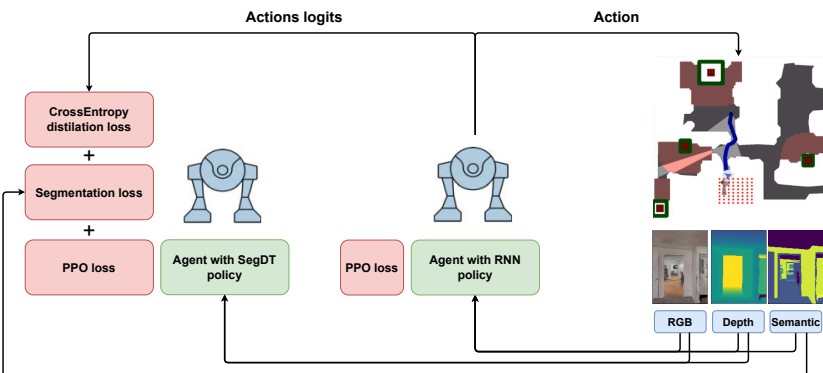

Figure 3: Diagram illustrating the fine-tuning of SegDT with trajectories from the RNN-based policy. Knowledge transfer from the RNN-based policy was achieved using cross-entropy loss. Additionally, segmentation loss was applied to SegDT, and PPO loss was utilized for both models.

For offline training of SegDT, we collect a dataset consisting of 16080 episodes in our 173 training scenes. The ground truth trajectories for behavioral cloning were obtained from the state-of-the-art RL algorithm for object goal navigation (Staroverov et al., 2023) using ground truth segmentation as input. The dataset for offline training contains 40 categories of the Matterport3D dataset (Chang et al., 2017) as goals for navigation, with the exception of 12 object categories: curtain, ceiling, column, door, floor, misc, objects, stairs, unlabeled, wall, window, and picture.

**Offline training.** We pre-train the Mask2Former segmentation model on a dataset consisting of 125K images collected in HM3DSem v0.2 training scenes with the same training parameters as in the original Mask2Former paper (Cheng et al., 2022). We render an image of size $160 \times 120$ in the Habitat environment and pad it to a square image resolution of $160 \times 160$, leaving the rest of the rendering parameters the same as in the Habitat Challenge 2023 (Yadav et al., 2023a). During offline training, we freeze the segmentation model. To train the remaining modules of SegDT, we use the AdamW (Loshchilov & Hutter, 2017) optimizer with a learning rate of $3 \times 10^{-4}$, $\beta_1 = 0.9$, $\beta_2 = 0.98$, $\lambda = 0.01$ and linear decay of learning rate. We use batch size equal to $8$ and a maximum of 64 frames from GT trajectories during training. The parameters of pretrained Mask2former are used to initialized parameters of segmentation modules of SegDT.

**Online fine-tuning.** As an RL algorithm, we use PPO with Generalized Advantage Estimation (Schulman et al., 2018). We set the discount factor $\gamma$ to 0.99 and the GAE parameter $\tau$ to 0.95. Each worker collects (up to) 64 frames of experience from 18 agents running in parallel (all in different scenes) and then performs 5 epochs of PPO. We use Adam (Kingma & Ba, 2017) with a learning rate of $1 \times 10^{-5}$. The agent receives terminal reward $r_T = 2.5$ SPL, and shaped reward $r_t(a_t, s_t) = -\Delta_{\text{geo\_dist}} - 0.01$, where $\Delta_{\text{geo\_dist}}$ is the change in geodesic distance to the goal by performing action $a_t$ in state $s_t$.

**Online validation.** To validate the agent strategy in the environment, we select a sample of 112 episodes on 9 validation scenes with 6 categories of objects from the Habitat Challenge (Yadav et al., 2023a) (bed (20 episodes from 112), toilet (20 episodes from 112), plant (20 episodes from 112), tv (20 episodes from 112), chair (20 episodes from 112), sofa (20 episodes from 112)). The agent can take up to 64 steps.

## 5.2 ONLINE SEGMENTATION QUALITY

**Baseline segmentation.** SegDT aggregates information from several previous frames to improve the segmentation quality for the current frame. Therefore, we compare the performance of the SegDT approach with the Single Frame Mask2Former (Cheng et al., 2022) baseline that makes predictions for the same frame sequence as SegDT. The Single Frame Mask2Former segments every frame in the sequence individually. We expect segmentation improvement for episodes where the agent frequently observes the target object. Such episodes mainly include episodes that ended with

Table 1: Comparison of the SegDT with other state-of-the-art methods for Object Goal Navigation task.

| Method | SR | SPL | SoftSPL |
|---|---|---|---|
| DD-PPO (500 steps) (Wijmans et al., 2020) | 10.2 | 2.1 | 14.6 |
| OnavRIM  (Chen et al., 2023) | 0.0 | 0.0 | 25.6 |
| OnavRIM (500 steps) (Chen et al., 2023) | 33.9 | 9.6 | 13.4 |
| PIRLNav  (Ramrakhya et al., 2023) | 25.7 | 24.3 | 43.1 |
| PIRLNav (500 steps)  (Ramrakhya et al., 2023) | 34.8 | 32.2 | 49.0 |
| RL with RNN and GT segmentation | 49.1 | 36.4 | 58.5 |
| **SegDT** with GT segmentation | **47.3** | **44.7** | **56.3** |
| RL with RNN and predicted segmentation | 31.2 | 28.2 | 46.2 |
| **SegDT** with predicted segmentation | **40.2** | **38.3** | **51.5** |

Table 2: Ablation of segmentation and navigation quality. We compute mIoU for two types of trajectories: Shortest Path Follower (SPF) trajectories and trajectories of successful episodes for each navigation method.

| Navigation semantics | Frame Sequence | mIoU (SPF trajectories) | mIoU (Success trajectories) | SR | SPL | SoftSPL |
|---|---|---|---|---|---|---|
| GT | Single Frame | – | – | 47.3 | 44.7 | 56.3 |
| Mask2Former | Single Frame | 51.8 | 59.2 | 38.0 | 36.2 | 49.9 |
| **SegDT** | **Navigation** | **53.7** | **70.4** | **40.2** | **38.3** | **51.5** |

success. Therefore, we evaluate the segmentation quality for two types of trajectories: shortest path trajectories for all 112 validation episodes and successful trajectories for each navigation algorithm. The shortest path trajectories were obtained from a classical planning algorithm  (Kumar et al., 2018). This planner greedily fits actions to follow the geodesic shortest path between the agent starting point and the goal position. For each step $t$, we consider as a baseline segmentation the Single Frame Mask2Former masks predicted for the input image $I_t$.

**Segmentation metric.** SegDT uses only target object masks to predict actions, so the navigation quality depends primarily on the quality of segmentation of these categories. For each episode, we compute the standard mean Intersection over Union ($mIoU$) (Jain et al., 2023) metric for four target categories: sofa, TV, armchair, plant, toilet and bed. We then average the resulting values across all successful episodes.

### 5.3 RESULTS

We compare the quality of our approach with state-of-the-art methods for object goal navigation. The comparison results are shown in Table 1. We use weight models trained to solve the task of navigation to a goal object. For each method, we use the action and observation spaces used in their training pipeline and limit trajectory length up to 64 steps if not stated otherwise.

The DD-PPO method(Wijmans et al., 2020) performs poorly with the goal reacher task since it relies only on the object goal class and does not cope well with semantic understanding of the scene. But in the task of navigating to a point, it shows results comparable to humans (Wijmans et al., 2020). OnavRIM (Chen et al., 2023) and PIRLNav  (Ramrakhya et al., 2023) are trained on human-collected trajectories. These trajectories have great length and usually start with an exploration of the environment. Therefore, OnavRIM and PIRLNav  (Ramrakhya et al., 2023) spend most of the time exploring the room and only then returning to the target, which in most cases exceeds the limit of 64 steps. Additionally, we present the navigation metrics for these methods with increasing the number of steps to 500 (the maximum episode length in the Habitat Challenge). As can be seen from Table 1, despite increasing the episode length, the OnavRIM (Chen et al., 2023) and PIRLNav  (Ramrakhya et al., 2023) methods show lower performance than SegDT. To avoid this effect, we employ a reward that penalizes the agent for deviating from the target when it is visible. We compare SegDT with the RNN-based GoalReacher skill (Staroverov et al., 2023) used for sampling trajectories during offline and online training. First, we use ground truth segmentation as input data. In this case, SegDT significantly outperforms GoalReacher in path efficiency, as shown by the SPL and SoftSPL metrics in Table 1. Then, we use predicted segmentation along with RGBD data. Here, the navigation

Table 3: Ablation of ground truth (GT) trajectories choice for Behavioral Cloning.

| GT trajectories source | SR | SPL | SoftSPL |
|---|---|---|---|
| Shortest Path Follower | 8.0 | 6.7 | 27.3 |
| **RNN-based GoalReacher skill (Staroverov et al., 2023)** | **18.0** | **16.3** | **33.9** |

quality of NN-based GoalReacher skill degrades significantly, while SegDT remains robust to noisy segmentation data due to segmentation loss during training.

We assess the impact of using previous frames to predict segmentation on segmentation and navigation quality. After training SegDT on offline and online data, we validate it in the environment using different segmentation masks to predict actions. We compare three segmentation methods: ground truth, SegDT, and the baseline Single Frame Mask2Former. Table 2 shows a slight decrease in navigation quality when switching from ground truth to SegDT-predicted segmentation. The baseline Mask2Former produces lower-quality masks, leading to a further decrease in navigation quality when used to predict actions.

Table 3 compares the impact of the selected source of ground truth trajectories on the quality of pretraining during offline Behavioral Cloning stage. Table 3 demonstrates that trajectories collected using the RNN-based GoalReacher skill (Staroverov et al., 2023) provide higher training quality on offline data compared to trajectories obtained from a classical planning algorithm (Kumar et al., 2018).

## 5.4 VISUALIZATION

Figure 4 demonstrates the qualitative effect of improving segmentation using SegDT for different categories of target objects. The main effect is expressed in filling segmentation gaps if the target object was present in previous frames. The aggregation of information from several frames improves the quality of instantaneous predicted mask contours.

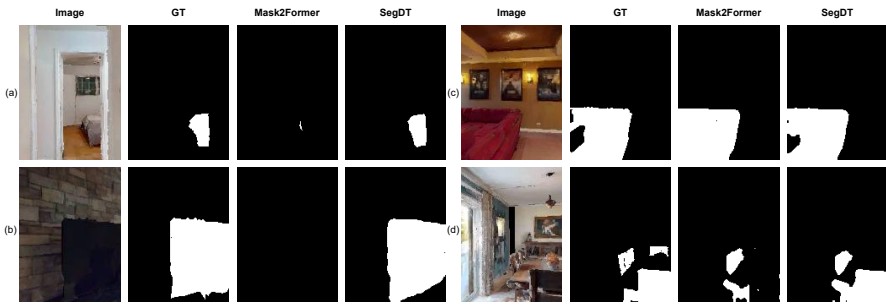

Figure 4: The segmentation results of SegDT compared to the baseline Mask2Former model.

## 6 CONCLUSION

Our results show that joint training of a multimodal decision transformer for segmentation and navigation improves the performance of both tasks. Two-phase training of the Segmenting Decision Transformer (SegDT) using additional training using DD-PPO in the environment can further improve the quality of navigation.

As a limitation of the proposed approach we can mention its computational complexity. The speed of inference slows down the fine-tuning of the action policy in the environment. Another limitation is the use of the pre-trained Mask2Former model to initialize the parameters of the segmentation modules of SegDT.

Another research direction is to create a method for selecting the most valuable frames for calculating the segmentation loss during training of SegDT.

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
