# OpenReview forum: "Semantic Object Navigation with Segmenting Decision Transformer"
_ICLR.cc/2025/Conference — ICLR 2025 Conference Withdrawn Submission_

### Official Review · Reviewer_N5W1 · 2024-10-28

**Soundness:** 2
**Presentation:** 3
**Contribution:** 1
**Rating:** 1
**Confidence:** 4

**Summary:**

This paper studies the problem of navigating an embodied agent to an object that is visible in the initial view of the agent. The authors contribute a method that predicts agent actions from a transformer that receives a long-horizon history of images, depth, and predicted object segmentation. The method is compared to RL-based methods for object search (i.e. the compared methods are originally designed to solve finding the object and then navigating to it), achieving better performance than these methods in an evaluation setup similar to the habitat object navigation challenge.

**Strengths:**

- I find the writing easy to follow and understand.
- The considered task itself is well defined and the experiments in the established habitat setting are straightforward and make sense.
- The introduction of an additional baseline with much more steps is a good idea to make the comparison less biased to these methods that are trained on both exploration and object nav.

**Weaknesses:**

- The biggest weakness of this paper to me is that I cannot see a valuable research question that this paper answers.
  - Object navigation is not a very challenging task and the authors actually use a simple geometric shortest-path method to generate training data. To investigate the best possible to solution for object navigation, experiments therefore would need to consider a much broader range of methods.
  - In the habitat challenge on the other hand this task is used as a benchmark for RL-based navigation methods (i.e. how to navigate an embodied agent based on images). That is what all the considered baselines here do, but the contributed method itself is not an RL based method, so it does not help us to advance understanding how to best do image-based RL for navigation.
  - the third possible research question that I see for this work is how to best do robot navigation planning with transformers, i.e. leveraging attention over long horizons. For this however I would expect a comparison to other transformer-based action planners such as [1, 2, 3].
- I am not convinced that the investigated problem of navigating to an object that is already in view is really useful or challenging. I think a naive solution of using a single-view depth map with some shape completion and an optimal, fast RRT* planner will solve this with a much higher success rate than the 40% of the proposed method.
- There are a couple of aspects that make the main evaluation experiments unfair:
  - The method specifically considers the task of moving to an object that is visible in the first view, while the compared methods are not built on that assumption. This makes the comparison slightly biased, because all other methods are likely to be based on a prior assumption that some exploration is necessary in the beginning. A more fair comparison would be to either retrain these prior methods on the same task/action distribution, or additionally compare by running any of these methods to explore until the object is predicted to be in view, and then switch between the prior method and the proposed method.
  - The proposed method is trained on 28 object categories (line 346). However, the evaluation considers only 6 categories in Table 1 (see line 366) and only 4 categories in Table 2 (see line 408).
  - In Table 2, mIoU for Success Trajectories: It is not clear from the text whether each row considers the same trajectories and these are trajectories successful with the proposed model (in this case there would be a sampling bias because it might be that these were successful because the segmentation model was better there), OR each row is evaluated on a different set of trajectories, making it impossible to compare because the used metric mIoU is very sensitive to object size, so it should be compared against the exact same ground truth. I think it would be best to compare here on the subset of trajectories that are successful regardless of the segmentation model.




[1] Shridhar, M., Manuelli, L., & Fox, D. (2023). Perceiver-Actor: A Multi-Task Transformer for Robotic Manipulation. Retrieved from https://proceedings.mlr.press/v205/shridhar23a.html
[2] Brohan, A., Brown, N., Carbajal, J., Chebotar, Y., Dabis, J., Finn, C., … Zitkovich, B. (2023). RT-1: Robotics Transformer for Real-World Control at Scale. Retrieved from https://arxiv.org/abs/2212.06817
[3] Driess, D., Xia, F., Sajjadi, M. S. M., Lynch, C., Chowdhery, A., Ichter, B., … Florence, P. (2023). PaLM-E: An Embodied Multimodal Language Model. Retrieved from https://arxiv.org/abs/2303.03378

**Questions:**

- lines 113-134: I find the comparison to mapping-based and next-best-view based methods unfair. In the first paragraph the argument against map-based methods is their memory consumption (actually in the considered problem where the whole action is limited to a 10m perimiter the map size is constant), and then the argument against next-best-view methods is that the proposed method can consider a longer history (i.e. larger memory).
- line 134: I don't think it is fair to say that this metod considers "complex photorealisitc scenes" if the images are at 120x160 resolution and rendererd from HM3D, which are textured, incomplete meshes and nothing like a photorealistic rendering.
- Section 4.2. It is unclear to me where the conceptual difference between Offline and Online training lays. It seems to me both are actually offline trainings where a motion policy is distilled from a RL-based method into the transformer model.
- line 311: Shouldn't the target rather be minimum amount of steps and highest success rate / lowest crash rate? Minimizing cumulative distance seems to me rather like a heuristic for a reward function but not a goal that is meaningful for the considered task.
- line 323: Here the authors say 10m away from the goal, in line 150 they say 7m. Which is correct?
- line 350: Why do you render 160x120 and then pad to 160x160 instead of rendering 160x160? This seems like a bad method design. Is this a fair data input for the other methods?
- line 352: Given that the segmentation model is frozen, what is actually trained here? Only the linear projections and learnable input tokens to GPT?
- Table 1: Following my thoughts on Weakness 1, why is the shortest path method not compared here?
- line 450: I am not sure whether the conclusion of the authors is correct. In my opinion, the ablation study says nothing about the quality of the training signal and rather something about the kind of data this particular model requires. E.g. presumably the RL-based motion will have the target object more often in view, while the viewing direction will not be important for the optimal geometric plan. Given that the proposed method relies on a history of camera views, the RL policy is favourable even if that means the proposed method does not learn the optimal trajectory.

**Details Of Ethics Concerns:**

This paper studies a widely established task (object goal navigation) in simulation and without human annotators.

---

### Official Review · Reviewer_fwiu · 2024-11-04

**Soundness:** 2
**Presentation:** 2
**Contribution:** 1
**Rating:** 3
**Confidence:** 5

**Summary:**

This paper addresses the challenge of reaching a target object in navigation tasks, focusing on the sensitivity to errors in semantic understanding of the scene. The authors propose SegDT, a method that enhances object navigation by integrating scene semantics within a multimodal and multitasking framework. The approach involves joint training of a segmentation model and a decision transformer model, allowing the aggregation of information from multiple multimodal frames for predicting actions and segmentation masks. The method includes a pre-training phase followed by online fine-tuning to address issues related to long-term credit assignment and sampling efficiency. Experiments were conducted in the Habitat Sim environment to evaluate the proposed method.

**Strengths:**

- Figures 2 and 3 clearly illustrate how SegDT works, which helps the reader gain a better understanding.

- The writing of the paper is clear and relatively easy to comprehend.

**Weaknesses:**

- The performance is relatively low. The authors report a performance of 40.2% on HM3D, while recent work such as PEANUT [1] has achieved 64% on the same dataset. This significant gap raises concerns about the effectiveness of the proposed method.
> [1] PEANUT: Predicting and Navigating to Unseen Targets. ICCV 2023: 10892-10901

- Recent zero-shot navigation approaches, such as VLFM [2], have achieved 52.5% on HM3D without requiring training. In contrast, the proposed method necessitates data collection and training, calling into question its validity.
> [2] VLFM: Vision-Language Frontier Maps for Zero-Shot Semantic Navigation. ICRA 2024: 42-48

- The experiments are conducted solely on HM3D, whereas related ObjectNav works typically evaluate their methods across multiple datasets, including AI2THOR, RoboTHOR, ProcTHOR, Gibson, MP3D, and HM3D.

- The related work section omits several important ObjectNav studies, particularly some modular-based approaches.
The visualizations in Figure 4 indicate that the segmentation results do not appear to be significantly better than the baseline.

**Questions:**

- I am somewhat confused about the meaning of Figure 1. It seems to explain the ObjectNav task, which has already been widely studied, making this figure somewhat redundant.

- I would like to know the rationale behind the choice of model architecture. Does the term "GPT architecture" refer to a decoder-only structure? It may be necessary to justify why such a structure is preferred over alternatives like encoder-only or encoder-decoder architectures.

- I have concerns regarding the generalization of this method, as modular-based approaches are also a significant consideration in ObjectNav. Furthermore, regarding the authors’ focus on the challenge of "reaching the found target," modular-based methods treat it as a point-to-point problem (with a success rate of 99%). I am curious about the generalizability of SegDT and whether it can be adapted to modular-based methods.

---

### Official Review · Reviewer_Lczn · 2024-11-09

**Soundness:** 3
**Presentation:** 2
**Contribution:** 2
**Rating:** 5
**Confidence:** 3

**Summary:**

This paper addresses a subtask of the object visual navigation problem where the agent learns a policy to reach a visible found object. The proposed method learns a semantic segmentation task in addition to the policy learning to improve the object reaching efficiency.

The policy is instantiated as a transformer-based model called SegDT. The training process of SegDT consists of several phases: (1). Pre-training the segmentation module on the HM3DSem dataset. (2). Training the SegDT policy with offline demonstrations by behavior cloning. (3). Train an additional RNN policy online with ground truth segmentation as input. (4). Fine-tuning SegDT policy on trajectories generated by the RNN policy to transfer the knowledge.

The experimental results in Habitat Sim environment show that the proposed approach outperforms the previous object navigation methods, achieving more efficient pathfinding to the given visible object goal.

**Strengths:**

This paper addresses the problem of improving an agent's ability to navigate to a nearby visual goal. The proposed method tackles this problem from a visual recognition perspective by explicitly training the agent to perform semantic segmentation in addition to navigation. The paper focuses on a specfic stage of the general object navigation task, where it assumes that the target object has already been found. In contrast, prior work primarily focuses on effciently locate the target object in an unknown environment, which is the main challenge in object navigation. Based on the assumption that the target object has already been located, the problem setting in this paper requires initializing the agent at a location where the target object is nearby (within 7 meters) and visible, which is a relatively constrained scenario and makes the contribution limited.

**Weaknesses:**

The main limitation of the paper is that it reduces the original object navigation problem to an easier problem by assuming that the target object has already been found. As a result, the problem addressed here is more accurately described as a goal-reaching task rather than a complete object navigation problem. This shift in focus may make the comparison to prior methods less fair. For example, when comparing to object navigation methods such as OnavRIM and PIRLNav in a goal-reaching setting, all methods should be initialized to random locations in the same range where the target object is visible. It is unclear whether this is ensured in the experiments. To demonstrate the advantages of the proposed method in solving the full object navigation problem, it may be helpful to show that incorporating the semantic segmentation task does not harm the success rate of locating the object in the exploration phase while still outperforming OnavRIM and PIRLNav. Other aspects to ensure a fair comparison include that the reward functions in different methods should rely on the same level of supervision. For example, the proposed method assumes access to the geodesic distance to the goal in reward computation, whereas the original PIRLNav paper only uses sparse success/failure rewards.

If the comparison is fair in the current experiment setup, the main observation is that slight performance improvement can be achieved in nearby object goal-reaching tasks when training with additional supervision—semantic segmentation ground truth—compared to methods without such supervision (DDPPO, OnavRIM, PIRLNav). The result is unsurprising, especially given that extra large computational complexity and cost are introduced.

The paper would benefit from a clearer presentation if the novelty and the reasons behind the design of the approach were highlighted, analyzed, and separated from training and implementation details. Additional ablation studies could also be conducted to clarify the importance of each design choice.

**Questions:**

In Table 1, why does RL with RNN and GT segmentation achieve better SR and SoftSPL than SegDT with GT segmentation?

---

### Official Review · Reviewer_hkUg · 2024-11-09

**Soundness:** 2
**Presentation:** 2
**Contribution:** 1
**Rating:** 3
**Confidence:** 3

**Summary:**

The paper addresses the challenging task of semantic object navigation (through object segmentation maps and reinforcement learning), in which an agent is placed randomly within an unknown environment and must navigate towards a specified object category. Towards this goal, the agent must first explore its environment to identify an instance of a given semantic goal and afterwards reach that object. The current submission introduces SegDT, a method that combines segmentation and action prediction in a single transformer model. The work focuses specifically on the "reaching" stage of object navigation, where an agent must navigate to a visible target object, without the need for exploration, simplifying its task considerably.
As contributions, SegDT simultaneously handles and improves semantic segmentation and action prediction through a knowledge transfer mechanism, where an RNN-based policy leverages ground truth segmentation to effectively teach a transformer-based policy that must learn to generate its semantic understanding. The practical value of these contributions is demonstrated through experimental validation in the Habitat environment (portraying indoor scenes).

**Strengths:**

The paper presents an interesting approach to combining semantic understanding with navigation in a unified transformer-based architecture. The use of a transformer to aggregate information from multiple frames for both segmentation and action prediction is novel and shows promise. The two-phase training procedure intelligently addresses the known limitations of transformers in RL settings, particularly their poor sample efficiency and credit assignment issues.

**Weaknesses:**

Several significant limitations raise concerns about the contributions of the current submission. First, the authors restrict themselves to the tackling only the navigation phase, assuming the target object is already visible. This is a major simplification of the real navigation problem and sidesteps many of the harder challenges in semantic navigation. From my understanding the methods in Table 1 tackle the full navigation problem including the exploration stage, which this method does not and because of favourable position / placement of the agent in the environment it is clear why it would be more effective and better performing.  The comparison feels somewhat unfair when the proposed method only handles a subset of the task.

Also, the results reported in Table 2 favours the proposed method since it considers multiple previous frames, which is important for navigation, having a better representation of temporal context, therefore I do not consider it to be a contribution, since it's a clear advantage over a baseline using a singular frame.

The overall presentation needs improvements both in terms of quality and clarity of the content, but also regarding the presentation of the main flow >> for instance Figure 1 is duplicated for no reason, without proper explanation of the elements neither in the text nor the caption, representing spatial information visually can we done more efficiently. Also, there is conflicting or confusing information within the body of the paper:
For instance, for the agent setup in
L150-151 - "Therefore, we initialized the agent at the random viewpoint of
the semantic goal at a maximum distance of seven meters" then
L322-323 - "... the target object is in the agent's field of view and the agent is no more than 10 meters away from the goal."

Other minor comments:
L441-442 - This statement can be improved. A suggestion for better clarity would be: "We evaluate how using previous frames for segmentation prediction impacts both segmentation accuracy and navigation performance."

**Questions:**

My questions relate to the main limitations, also highlighted by the authors in the paper:

1. The authors mentioned the computation complexity as one of the limitations but failed to provide concrete numerical evidence to back this up. How does the computational cost of SegDT compare to existing methods, for instance, the ones in Table 1, particularly during inference?

2. How could the approach be extended to handle the full navigation problem, including exploration and would then compare to the methods in Table 1?

3. Did the authors explore the sensitivity of their method to the quality of the initial pre-trained Mask2Former baseline?

4. Why only a sequence length of 64 frames? Have the authors explored other values? What is the impact of the sequence length on both performance and computational requirements?

---

### Official Review · Reviewer_qLAi · 2024-11-11

**Soundness:** 3
**Presentation:** 3
**Contribution:** 3
**Rating:** 6
**Confidence:** 3

**Summary:**

The authors propose SegDT (Segmenting Decision Transformer) -- a novel approach for semantic object navigation that combines segmentation and navigation in a single transformer model.

The key contribution is jointly training a segmentation model with a decision transformer to both improve semantic understanding and navigation capabilities. The method uses a two-phase training process- a pre-training step using collected trajectories and an online policy fine-tuning using PPO algorithm with knowledge transfer from an RNN-based policy using ground truth segmentation and transfers its knowledge to the proposed transformer-based model, which trains a better segmentation model.

The model achieved competitive results in the Habitat Sim environment for object goal navigation.

**Strengths:**

### Novelty
- Novel technical approach combining segmentation and navigation in a single transformer model
- Clever training strategy that addresses transformer limitations through RNN knowledge transfer

### Performance
- Competitive results over state-of-the-art methods for object navigation

### Component reasoning
- Clear ablation studies
- Good qualitative results showing improved segmentation through temporal information

**Weaknesses:**

### Explainability
- Limited analysis of why the approach works better - could benefit from more insight into what the transformer is learning

### Performance
- Higher computational complexity during inference
- Reliance on pre-trained Mask2Former model for initialization
- Evaluation focused on relatively short trajectories (64 steps) compared to some baselines that use 500 steps
- Tests on a synthetic dataset only

**Questions:**

1. How does the performance scale with longer trajectories (> 64 steps)?
2. Why not compare with a 2D-to-3D lifting method such as [1*] to determine whether the improved semantic segmentation is related in any way with the policy?
2. The paper mentions selecting "the most valuable frames for calculating segmentation loss" as future work - what criteria would you propose for this selection?
___
[1*]Yan, M., Zhang, J., Zhu, Y., & Wang, H. (2024). Maskclustering: View consensus based mask graph clustering for open-vocabulary 3d instance segmentation. In Proceedings of the IEEE/CVF Conference on Computer Vision and Pattern Recognition (pp. 28274-28284).

---

### Note · Authors · 2024-11-15

I have read and agree with the venue's withdrawal policy on behalf of myself and my co-authors.